# Tool Unlearning for Tool-Augmented LLMs

**Jiali Cheng** [1]   **Hadi Amiri** [1]

## Abstract

Tool-augmented large language models (LLMs) are often trained on datasets of query-response pairs, which embed the ability to use tools or APIs directly into the parametric knowledge of LLMs. As these models are increasingly deployed in real-world applications, there is a need for them to forget specific tools–for example, due to security vulnerabilities, privacy regulations, or tool deprecation. This work presents "tool unlearning" as a novel machine unlearning task that presents distinct challenges beyond traditional sample-level unlearning: it requires removing functional knowledge rather than individual data points, managing the high cost of LLM optimization, and developing principled evaluation metrics. To address these challenges, we propose TOOLDELETE, the first unlearning framework designed specifically for tool-augmented LLMs. It implements three key properties for effective tool unlearning and introduces a new membership inference attack (MIA) model for effective evaluation. Extensive experiments on multiple tool learning datasets and tool-augmented LLMs show that TOOLDELETE effectively unlearns both randomly selected and class-specific tools, while preserving knowledge on remaining tools and maintaining performance on general tasks.

## 1. Introduction

Tool-augmented Large Language Models (LLMs) can use external tools such as calculators (Schick et al., 2023), Python interpretors (Gao et al., 2023), APIs (Tang et al., 2023), or AI models (Patil et al., 2023) to complement the parametric knowledge of vanilla LLMs and enable them to solve more complex tasks (Schick et al., 2023; Patil et al., 2023). They are often trained on query-response pairs, which embed the ability to use tools *directly* into model parameters.

Despite the growing adoption of tool-augmented LLMs, the ability to selectively unlearn tools has not been investigated. In real-world applications, tool unlearning is essential for addressing critical concerns such as security, privacy, and model reliability. For example, consider a tool-augmented LLM deployed in a healthcare system and trained to use APIs for handling patient data. If one of the APIs is later flagged as insecure due to a vulnerability that could expose sensitive information and violate regulations like HIPAA, tool unlearning is necessary to ensure that the LLM can no longer invoke the insecure API. Similarly, when tools undergo major updates, such as the Python transformers package moving from version 3 to version 4, tool unlearning becomes essential to prevent the LLM from generating outdated or erroneous code. The goal of this work is to address this gap by investigating tool unlearning and providing a solution for this crucial task.

We introduce and formalize the new task of **Tool Unlearning**, which aims to remove the ability of using specific tools from a tool-augmented LLM while preserving its ability to use other tools and perform general tasks of LLMs such as coherent text generation. Ideally, an effective tool unlearning model should behave as if it had never learned the tools marked for unlearning. Tool unlearning fundamentally differs from traditional sample-level unlearning as it focuses on removing "skills" or the ability to use specific tools, rather than removing individual data samples from a model. In addition, success in tool unlearning should be measured by the model's ability to forget or retain tool-related skills, which differs from traditional metrics such as measuring likelihood of extracting training data in sample-level unlearning. These differences are discussed in detail in §2.

Removing skills requires modifying the parameters of LLMs, a process that is computationally expensive and can lead to unforeseen behaviors (Cohen et al., 2024; Gu et al., 2024). In addition, existing membership inference attack (MIA) techniques, a common evaluation method in machine unlearning to determine whether specific data samples were part of training data, are inadequate for evaluating tool unlearning because they focus on sample-level data rather than tool-based knowledge.

---

[1]University of Massachusetts Lowell, USA. Correspondence to: Jiali Cheng <jiali_cheng@uml.edu>, Hadi Amiri <hadi_amiri@uml.edu>.

*Proceedings of the 42nd International Conference on Machine Learning*, Vancouver, Canada. PMLR 267, 2025. Copyright 2025 by the author(s).

To address these challenges, we propose TOOLDELETE, the first tool unlearning algorithm for tool-augmented LLMs, which satisfies three key properties for effective tool unlearning: *tool knowledge removal*, which focuses on removing any knowledge gained on tools marked for unlearning; *tool knowledge retention*, which focuses on preserving the knowledge gained on other remaining tools; and *general capability retention*, which maintains LLM's general capability on a range of general tasks such as text and code generation using ideas from task arithmetic (Ilharco et al., 2023; Bărbulescu & Triantafillou, 2024). In addition, we develop LiRA-Tool, an adaptation of the Likelihood Ratio Attack (LiRA) (Carlini et al., 2022; Pawelczyk et al., 2024) to tool unlearning, to assess whether tool-related knowledge has been successfully unlearned. Our contributions are:

- introducing and conceptualizing tool unlearning for tool-augmented LLMs,

- TOOLDELETE, which implements three key properties for effective tool unlearning;

- LiRA-Tool, which is the first membership inference attack (MIA) for tool unlearning.

Extensive experiments on multiple datasets and tool-augmented LLMs show that TOOLDELETE outperforms existing general and LLM-specific unlearning algorithms by $12.5$ and $9.1$ in accuracy on forget tools and retain tools respectively. In addition, it can save 74.8% of training time compared to retraining, handle sequential unlearning requests, and retain 95% performance in low resource setting.

## 2. Tool Unlearning: Preliminaries

To understand tool unlearning, we first introduce the concept of "tool learning," see Figure 1(a). Let $\mathcal{D} = \{\mathcal{T}, \mathcal{Q}, \mathcal{Y}\}$ be a dataset with $N$ tools $\mathcal{T}$, and $(\mathcal{Q}, \mathcal{Y})$ denotes query-output examples that demonstrate how to use the tools in $\mathcal{T}$. Each tool $t_i \in \mathcal{T}$ may have one or more demonstrations $\{\mathcal{Q}_i, \mathcal{Y}_i\}$, $|\mathcal{Q}_i| = |\mathcal{Y}_i| \geq 1$. Starting with an instruction-tuned LLM $f_0$, a tool learning algorithm explicitly trains $f_0$ on $\mathcal{D}$ and results in a *tool-augmented* model $f$ capable of using the $N$ tools in $\mathcal{T}$. We note that prior to explicit tool learning, the LLM $f_0$ may already have some tool-using capabilities such as performing basic arithmetic operations.

**Problem Definition:** Tool unlearning aims to remove specific tools from tool-augmented LLMs. Let $\mathcal{D}_f = \{\mathcal{T}_f, \mathcal{Q}_f, \mathcal{Y}_f\}$ denotes $k < N$ tools and their corresponding demonstrations to be unlearned from the tool-augmented model $f$, and $\mathcal{D}_r = \mathcal{D} \setminus \mathcal{D}_f = \{\mathcal{T}_r, \mathcal{Q}_r, \mathcal{Y}_r\}$ denotes the remaining tools and their demonstrations to retain. The goal is to obtain an unlearned model $f'$ that has limited knowledge on using $\mathcal{T}_f$ tools–can no longer perform tasks involving $\mathcal{T}_f$ tools–while preserving $f$'s ability to use $\mathcal{T}_r$ tools as before.

**Use Cases of Tool Unlearning** The ability to forget learned tools is essential in real-world applications. For example, addressing the insecure tools from untrustworthy developers that could be exploited by adversarial attackers; removing tools restricted by their providers due to copyright or privacy concerns, such as APIs that start allowing unauthorized downloads of book chapters or releasing publications that users did not author; unlearning broken or deprecated tool that lead to failed operations or corrupted outputs; unlearning tools that may no longer be needed; and managing limited model capacity, where new versions of tools necessitate replacing outdated ones. More examples of parameter-level tool unlearning are provided in Appendix A.

**Difference to Standard Unlearning Tasks** Tool unlearning is different from sample-level unlearning as it focuses on removing "skills" rather than individual training samples. **Objective**: sample-level unlearning aims to reduce the memorization likelihood or extraction probabilities of specific data samples $(q_i, y_i)$ (Jang et al., 2023), which is useful for removing copyrighted or private information. In contrast, tool unlearning targets the "ability" to solve tasks using tools marked for unlearning ($T_f$). For example, generating $f'(q_i)$ that is superficially different from $y_i$ (while preserving the semantics) is considered successful for sample-level unlearning. However, for tool unlearning, preserving skills and semantics indicate maintained knowledge on $T_f$, which makes unlearning a failure. Figure 1b shows successful tool unlearning, where the ability to use the API is forgotten, despite the high lexical memorization between output of the unlearned model and the training data. In addition, selectively removing knowledge from tool-augmented models is a challenging tasks because changes to one tool may unexpectedly affect the model's ability to use other tools–referred to as *ripple effect* in fact editing literature (Cohen et al., 2024; Gu et al., 2024). Furthermore, LLMs are general models that can conduct a wide range of tasks beyond tool using, and this ability must be retained. **Evaluation**: metrics like sequence extraction likelihood and perplexity are standard in sample-level unlearning. For tool unlearning, success is measured by the ability to forget or retain tool-related skills, which is more appropriate. **Data**: sample-level unlearning require access to all individual samples marked for unlearning, while tool unlearning does not. This aligns with "concept erasure" in diffusion models (Gandikota et al., 2023; Kumari et al., 2023) and zero-shot unlearning (Chundawat et al., 2023) but differs from traditional LLM unlearning (Yao et al., 2024). Later we demonstrate this in § 5.

**Importance of Parameter-Level Tool Unlearning** We observe that one can naively block tools at the prompt-level or remove tools from the tool set without updating the LLM. However, these shortcut solutions are insufficient to remove tool knowledge. *Firstly*, the knowledge on $\mathcal{T}_f$ persists in

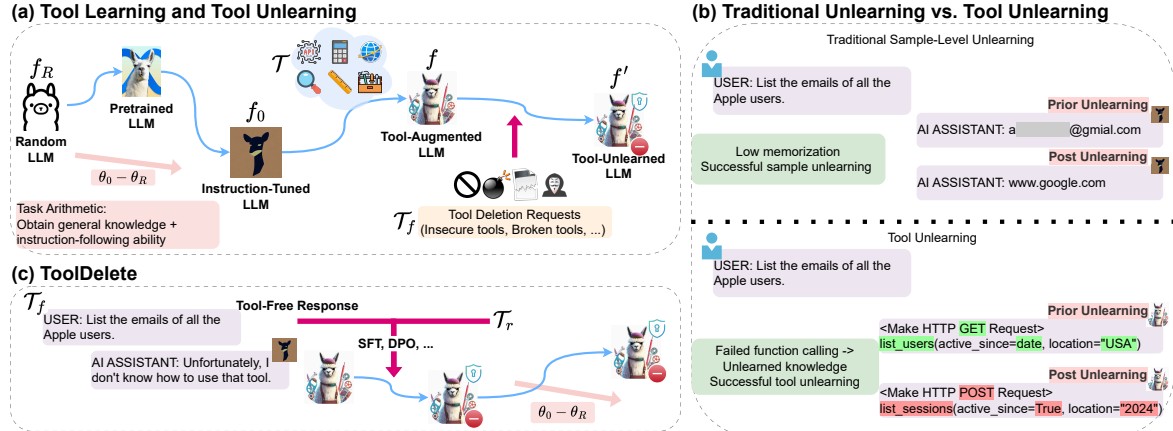

*Figure 1.* Tool Unlearning and the proposed TOOLDELETE approach. **(a)**: Illustration of tool learning and tool unlearning. Learned tools may be requested to be unlearned due to many reasons, such as tools being insecure, restricted, or deprecated. **(b)**: Differences between tool unlearning and traditional sample unlearning, in terms of objective and training data. **(c)**: Proposed method TOOLDELETE. We encourage the unlearned model $f'$ to follow the tool-free LLM $f_0$ which has never seen $T_f$ before. Meanwhile, we maintain its ability on $T_r$ by matching the capabilities of tool-augmented model $f$ through task arithmetic.

the parameters of $f'$, leaving the LLM still under threat. Adversarial agents / attackers can exploit this knowledge, which also bypasses prompt-level restrictions. Since existing LLMs do not guarantee 100% adherence to instructions or contextual information (Zhou et al., 2023; Zeng et al., 2024), they may ignore the tool set provided in the prompt and answer queries with their parametric knowledge (Goyal et al., 2023). In addition, tool unlearning at prompt level can create conflicts between the model's parametric knowledge and contextual information. This may lead to misinformation, hallucination, and other unpredictable behavior (Xu et al., 2024). Finally, we show in the experiments that prompt-level tool unlearning is indeed insufficient, see Table 1 (ICLU model), which aligns with existing works on LLM unlearning, where parameter update is required (Jia et al., 2024; Zhang et al., 2024b).

## 3. TOOLDELETE

We develop TOOLDELETE–an effective tool unlearning approach that removes the capability of using tools marked for unlearning ($\mathcal{T}_f$) or solving tasks that depend on them, while preserving the ability of using the remaining tools ($\mathcal{T}_r$) and performing general tasks such as text and code generation. TOOLDELETE implements three key properties for effective tool unlearning:

### 3.1. Tool Knowledge Deletion

Unlearning requires completely removing the knowledge of $\mathcal{T}_f$ that $f$ gained during tool learning, ideally as if $\mathcal{T}_f$ had never been part of the training set. In other words, knowledge about $\mathcal{T}_f$ is successfully removed if

the unlearned model $f'$ has no more knowledge than the tool-free model $f_0$ about $\mathcal{T}_f$.

**Definition 3.1** (Tool Knowledge Deletion (TKD)). Let $t_i \in \mathcal{T}_f$ denote a tool to be unlearned and $g$ be a function that quantifies the amount of knowledge a model has about a tool. The unlearned model $f'$ satisfies tool knowledge deletion if:

$$\mathbb{E}_{t_i \in \mathcal{T}_f} [g(f_0, t_i) - g(f', t_i)] \geq 0. \tag{1}$$

This formulation allows users to control the extent of knowledge removal from $f'$. For instance, when we unlearn a "malicious" tool that calls a malignant program, we may require $f'$ retains no knowledge of this tool, i.e. $g(f', t_i) = 0$. In less critical cases, users can choose to reset $f'$'s knowledge to *pre*-tool augmentation level, i.e. $g(f', t_i) = g(f_0, t_i)$

To measure tool knowledge in LLMs, we follow previous works that used prompting to probe LLMs' knowledge (Brown et al., 2020; Singhal et al., 2023), i.e. adopting the output of LLMs as their knowledge on a given tool. For each $t_i \in \mathcal{T}_f$ and its associated demonstrations $\{\mathcal{Q}_i, \mathcal{Y}_i\}$, we query the tool-free LLM $f_0$ with $\mathcal{Q}_i$ and collect its responses $\mathcal{Y}'_i = f_0(Q_i)$. Since $f_0$ has never seen $t_i$ or $\{\mathcal{Q}_i, \mathcal{Y}_i\}$, $\mathcal{Y}'_i$ represents the **tool-free response**. We then constrain the unlearned model $f'$ to generate responses similar to $\mathcal{Y}'_i$ to prevent it from retaining knowledge of $t_i$.

### 3.2. Tool Knowledge Retention

The unlearning process should preserve model's knowledge of tools in $T_r$. Ideally, all knowledge gained on $T_r$ during tool learning should be retained after unlearning.

**Definition 3.2** (Tool Knowledge Retention (TKR)). Let $t_m \in T_r$ denote a retained tool, and let $g$ be a function that

quantifies the amount of knowledge a model has about a tool. The unlearned model $f'$ satisfies tool knowledge retention if:

$$\mathbb{E}_{t_m \in \mathcal{T}_r}[g(f, t_m) - g(f', t_m)] = \epsilon, \qquad (2)$$

where $\epsilon$ is an infinitesimal constant, so that $f'$ retains the same knowledge of tools in $T_r$ as the original model $f$.

For effective tool knowledge retention, $f'$ is further fine-tuned using demonstrations associated with $\mathcal{T}_r$, or, more practically, a subset of $\mathcal{T}_r$ proportional to $\mathcal{T}_f$ for efficiency.

### 3.3. General Capability Retention via Task Arithmetic

Optimizing the above objectives can lead to effective unlearning, but it may not be sufficient to maintain the general capabilities of the unlearned model $f'$. As a foundation model, $f'$ is expected to retain abilities such as text and code generation, question answering, instruction-following, and basic mathematical reasoning. These capabilities either existed in $f_0$ prior to tool augmentation or do not depend on specific tools. Therefore, preserving the general capabilities of $f'$ is essential to guarantee that tool unlearning does not compromise the overall functionality of the model.

**Definition 3.3** (General Capability Retention (GCR)). Let $\mathcal{T}_G$ denote the general tasks used to evaluate LLMs. The unlearned model $f'$ satisfies general capability retention if it preserves the knowledge on $T_G$ that it originally obtained prior to tool learning:

$$\mathbb{E}_{t_g \in \mathcal{T}_G}[g(f_0, t_g) - g(f', t_g)] = \epsilon, \qquad (3)$$

where $\epsilon$ is an infinitesimal constant.

We propose to use task arithmetic (Ilharco et al., 2023; Bărbulescu & Triantafillou, 2024) as an efficient and effective approach to preserving the general capabilities of the unlearned model. Our objective is that $f'$ retains as much general knowledge as $f_0$, the instruction tuned LLM trained from a randomly initialized model $f_R$. Let $\theta_0$ and $\theta_R$ denote the parameters of $f_0$ and $f_R$ respectively. The difference vector $\theta_0 - \theta_R$ captures the direction of general knowledge acquisition. We apply this adjustment to $\theta'$ (the parameters of $f'$) to preserve its general knowledge:

$$\theta'^* \leftarrow \theta' + (\theta_0 - \theta_R). \qquad (4)$$

**Why Task Arithmetic?** Task arithmetic is efficient, practical, effective for preserving general capabilities (Ilharco et al., 2023; Bărbulescu & Triantafillou, 2024): **Efficiency**: the vector operation does not scale with dataset size, making it significantly more efficient than retraining on large datasets. **Practicality**: general capabilities obtained from pre-training and instruction tuning (Zhou et al., 2024) are

often impractical to replicate due to the size and limited availability of data–even in some open-source LLMs (Touvron et al., 2023b), the actual pre-training data is not fully open-source. In addition, reintroducing general knowledge from alternative datasets can lead to data imbalances and distributional biases. **Effectiveness**: applying $\theta_0 - \theta_R$ largely restores the foundational abilities of $f'$, such as text generation and instruction-following, without requiring expensive and time-consuming retraining on large datasets.

### 3.4. Training Details

To obtain the unlearned model $f'$, we solve:

$$\theta'^* = \arg\min_{\theta'} \underbrace{\mathbb{E}_{t_i \in \mathcal{T}_f}[g(f_0, t_i) - g(f', t_i)]}_{\text{knowledge deletion of } \mathcal{T}_f} +$$
$$\underbrace{\mathbb{E}_{t_m \in \mathcal{T}_r}[g(f, t_m) - g(f', t_m)]}_{\text{knowledge retention of } \mathcal{T}_r}, \quad (5)$$

and once the optimized model parameters $\theta'^*$ are obtained, we apply task arithmetic to reinforce general capabilities:

$$\theta'^* = \underbrace{\theta'^*}_{\text{post-optimization weights}} + \underbrace{\alpha(\theta_0 - \theta_R)}_{\text{knowledge retention of } \mathcal{T}_G}, \quad (6)$$

where $\alpha$ is a hyperparameter to control the magnitude of task arithmetic. The above formulation provides flexibility in training TOOLDELETE using various existing paradigms, including supervised fine-tuning (SFT) (Taori et al., 2023), direct preference optimization (DPO) (Rafailov et al., 2023), reinforcement learning from human feedback (RLHF) (Ouyang et al., 2022), parameter-efficient fine-tuning (PEFT) (He et al., 2022; Su et al., 2023), or quantization (Dettmers et al., 2022; Ma et al., 2024) techniques. Below we describe two variants of TOOLDELETE:

- **TOOLDELETE-SFT** fine-tunes $f$ using language modeling loss. On forget tools $\mathcal{T}_f$, we replace the original responses $\mathcal{Y}_f$ with tool-free responses $\mathcal{Y}'_f$. The samples for $\mathcal{T}_r$ are not modified.

- **TOOLDELETE-DPO** uses direct preference optimization (DPO) to prioritize wining responses over losing responses. For $(t_i, \mathcal{Q}_i, \mathcal{Y}_i) \in \mathcal{T}_f$ to be unlearned, we prioritize the corresponding tool-free response $\mathcal{Y}'_i$ over the original response $\mathcal{Y}_i$. For $(t_j, \mathcal{Q}_j, \mathcal{Y}_j) \in \mathcal{T}_r$, the original response $\mathcal{Y}_j$ is prioritized over the tool-free response $\mathcal{Y}'_i$.

### 3.5. LiRA-Tool for Tool Unlearning Evaluation

**Challenge** A key challenge in evaluating tool unlearning is the lack of membership inference attack (MIA) models to determine whether a tool has been truly unlearned. Existing MIA models typically evaluate individual training samples

by analyzing model loss, which is insufficient for tool unlearning. Unlike sample-level unlearning, tool unlearning focuses on removing abstract parametric knowledge of tools in $\mathcal{T}_f$, not just forgetting specific training samples. The key limitation of sample-based MIA is that the prompt-response pairs $(\mathcal{Q}_f, \mathcal{Y}_f)$ in the training set may not fully represent all aspects of a tool's functionality. As a result, sample-level MIA may "overfit" to a limited subset of tool related prompts and fail to holistically assess whether the tool-usage capability have been fully removed from the model's parametric knowledge (Lynch et al., 2024; Łucki et al., 2025; Hu et al., 2025).

**Solution**    To address the above limitation, we introduce "shadow samples", a diverse set of prompt-response pairs to probe various aspects of tool knowledge. We prompt GPT4 with different combinations of in-context examples to obtain a comprehensive set of prompt-response pairs with various prompt format, intention, and difficulty requirements. These samples will be used to stress-test the unlearned LLM $f'$ beyond the specific training prompts. This approach prevents overfitting to the original training data and provides a more reliable evaluation of whether the tool has truly been forgotten. To implement this, we extend Likelihood Ratio Attack (LiRA) (Carlini et al., 2022), the state-of-the-art MIA approach, to tool unlearning.

**Sample-level LiRA**    LiRA infers the membership of a sample $(x, y)$ by constructing two distributions of model losses: $\tilde{\mathbb{Q}}_{\text{in}}$ and $\tilde{\mathbb{Q}}_{\text{out}}$ with $(x, y)$ in and out of the model training set respectively. These distributions are approximated as Gaussians, with their parameters estimated based on "shadow models" trained on different subsets of the training data. The Likelihood-Ratio Test (Vuong, 1989; Carlini et al., 2022) is then used to determine whether $(x, y)$ is more likely to belong to $\tilde{\mathbb{Q}}_{\text{in}}$ or $\tilde{\mathbb{Q}}_{\text{out}}$. For LLMs, the test statistic is given by (Pawelczyk et al., 2024) as:

$$\Lambda = \frac{P\Big(l\big(f(x), y\big)|\tilde{\mathbb{Q}}_{\text{in}}\Big)}{P\Big(l\big(f(x), y\big)|\tilde{\mathbb{Q}}_{\text{out}}\Big)} = \frac{\Pi_{(x_i, y_i)\in\mathcal{D}_f} P_U\Big(l\big(f'(x_i), y_i\big)\Big)}{\Pi_{(x_i, y_i)\in\mathcal{D}_f} P_{T_r}\Big(l\big(f(x_i), y_i\big)\Big)}.$$
$$(7)$$

This approach, however, is insufficient for tool unlearning because it only assesses membership of specific training samples rather than measuring whether the model still retains the capability to use a tool.

**LiRA-Tool: Knowledge-level LiRA**    A major limitation of sample-level LiRA is in its reliance on training-set observations, which may not fully capture the knowledge distribution of an entire tool. Therefore, applying LiRA to tool unlearning can lead to overfitting to a specific subset of training prompts and failing to comprehensively assess whether the tool knowledge has been removed. We address this issue by

introducing LiRA-Tool. Instead of relying on observed training samples, we construct a "shadow distribution" $\mathbb{P}$ that generates tool-related query-response pairs. This allows us to sample diverse tool-specific prompts that test the model's ability to use the tool. The new likelihood-ratio test is:

$$\Lambda = \frac{\Pi_{t_i\in\mathcal{T}_f}\Pi_{(x,y)\in\mathbb{P}_{t_i}} P_U\Big(l\big(f'(x), y\big)\Big)}{\Pi_{t_j\in T_r}\Pi_{(x,y)\in\mathbb{P}_{t_j}} P_{\mathcal{T}_r}\Big(l\big(f(x), y\big)\Big)},$$
$$(8)$$

where $\mathbb{P}_{t_i}$ represents the shadow distribution for generating tool-learning samples for tool $t_i$. $P_U(\cdot)$ indicates the distribution of unlearned tools $T_f$ under the unlearned model $f'$, while $P_{T_r}(\cdot)$ denotes the distribution of the retain tools $T_r$ under the retained model $f$. In practice, we use GPT-4 to generate diverse shadow samples by prompting it with various distinct instructions to ensure that the evaluation set captures more comprehensive aspects of tool knowledge than the training set. Appendix E provides more details.

**Novelty of LiRA-Tool**    The key novelty in LiRA-Tool in the sue of "shadow samples," which introduce diversity across multiple dimensions. By moving beyond limited training prompts, LiRA-Tool ensures that the model loss reflect overall tool-using ability, rather than just sample-level memorization. Our loss-ratio formulation shares similarities to previous MIAs for sample-level unlearning, such as probability distribution comparison prior- and post-unlearning (Cheng et al., 2023; Cheng & Amiri, 2024a) and other adaptations of LiRA using shadow models (Kurmanji et al., 2023; Pawelczyk et al., 2024). However, to the best of our knowledge, this work is the first adaptation of LiRA for detecting tool presence in tool-augmented LLMs.

**Limitations of LiRA-Tool**    Shadow samples obtained from GPT-4 may not fully represent the complexity of the original tool-learning data and can potentially lead to incomplete approximations of the true knowledge distribution. However, despite this limitation, shadow samples provide a more comprehensive and consistent evaluation of a model's tool-using abilities compared to relying merely on observed training samples, which are often limited and incomplete. Expanding the diversity and robustness of shadow sample generation is indeed an important direction for future work.

## 4. Experimental Setup

**Datasets & Tool-Augmented LLMs**    We experiment with the following datasets and their corresponding LLMs:

- **ToolAlpaca** (Tang et al., 2023) is an agent-generated tool learning dataset consisting of 495 tools and 3975 training examples. **ToolAlpaca 7B** is fine-tuned on ToolAlpaca using Vicuna-v1.3 (Zheng et al., 2023).

*Table 1.* Tool unlearning performances when deleting 20% of tools on ToolAlpaca. Best and second-best performances are **bold** and underlined respectively. *Original* is provided *for reference only*. Results on other LLMs are shown in Appendix Table 5-6.

| | METHOD | $\mathcal{T}_t(\uparrow)$ | $\mathcal{T}_r(\uparrow)$ | $\mathcal{T}_f(\downarrow)$ | GENERAL CAPABILITY $\mathcal{T}_G(\uparrow)$ | | | | |
|---|---|---|---|---|---|---|---|---|---|
| | | | | | STEM | REASON | INS-FOLLOW | FACT | AVG. |
| | ORIGINAL (REF ONLY) | 60.0 | 73.1 | 75.7 | 31.7 | 17.1 | 22.6 | 25.0 | 24.1 |
| GENERAL | RETRAIN | 52.1 | 71.8 | 38.5 | 30.5 | 16.1 | 14.2 | 24.7 | 21.3 |
| | GRADASCENT | 33.3 | 51.4 | 34.6 | 21.4 | 10.4 | 12.9 | 13.1 | 14.5 |
| | RANDLABEL | 50.3 | 70.3 | 37.5 | 26.3 | 16.4 | 13.6 | 25.1 | 20.3 |
| | SALUN | 46.2 | 54.3 | 38.2 | 27.1 | 17.0 | 17.4 | 19.5 | 20.2 |
| LLM-SPECIFIC | ICUL | 49.1 | 74.8 | 58.3 | 12.4 | 8.7 | 1.6 | 6.2 | 7.3 |
| | SGA | 43.5 | 63.0 | 42.1 | 21.5 | 11.6 | 17.0 | 14.7 | 16.2 |
| | TAU | 43.8 | 61.7 | 42.5 | 22.0 | 17.6 | 22.3 | 21.7 | 20.9 |
| | CUT | 44.7 | 61.5 | 40.2 | 21.6 | 14.8 | 20.8 | 16.4 | 18.4 |
| | NPO | 50.8 | 66.9 | 30.1 | 20.7 | 15.3 | 21.9 | 18.9 | 19.2 |
| | SOUL-GRADDIFF | 50.4 | 68.3 | 33.8 | 31.6 | 17.2 | 21.4 | 20.8 | 22.7 |
| OURS | TOOLDELETE-SFT | 52.7 | 72.1 | 30.5 | 31.3 | 17.5 | 21.7 | 24.1 | **23.6** |
| | TOOLDELETE-DPO | **53.4** | **75.1** | **28.7** | 31.6 | 16.8 | 20.4 | 23.5 | 23.1 |

- **ToolBench** (Qin et al., 2024) consists of more than 16k real world APIs from 49 categories, where each training demonstration involves complex task solving traces. **ToolLLaMA** is fine-tuned on ToolBench using LLaMA-2 7B (Touvron et al., 2023b).

- **API-Bench** (Patil et al., 2023) focus on APIs that load machine learning models. **Gorilla** is fine-tuned on API-Bench from LLaMA 7B (Touvron et al., 2023a).

**Setup & Evaluation** We use the public checkpoints of the above tool-augmented LLMs as original models–the starting point for unlearning. Then we conduct unlearning experiments with 2–20% tools randomly selected as $\mathcal{T}_f$. We evaluate tool unlearning effectiveness, general capability of tool-unlearned LLMs, and robustness to membership inference attack (MIA). For **unlearning effectiveness**, we measure performance on test sets ($\mathcal{T}_T, \uparrow$), forget set ($\mathcal{T}_f, \downarrow$), and remaining set ($\mathcal{T}_r, \uparrow$), where "performance" reflects the ability to solve tasks that depend on specific tools, depending on the unique metrics in the original tool-augmented models $f$. For **general capabilities**, we evaluate the unlearned LLMs on a wide range of tasks: college STEM knowledge with MMLU (Hendrycks et al., 2021), reasoning ability with BBH-Hard (Suzgun et al., 2023), instruction-following with IFEval (Zhou et al., 2023), and factual knowledge with MMLU (Hendrycks et al., 2021). For **MIA**, we use the proposed LiRA-Tool; following prior work on LiRA (Pawelczyk et al., 2024), we train the shadow models with forget set size of {1, 5, 10, 20} and primarily evaluate the True Positive Rate (TPR) at low False Positive Rate (FPR) (TPR @ FPR = 0.01), where TPR means the attacker successfully detects a tool is present. Therefore, a lower TPR indicates better performance (privacy).

**Baselines** As there are no prior works on tool unlearning, we adapt the following unlearning methods to tool unlearning setting (see Appendix B for descriptions of the baselines): general unlearning approaches, including **GRADASCENT** (Golatkar et al., 2020; Yao et al., 2024), **RANDLABEL** (Graves et al., 2021), and **SALUN** (Fan et al., 2024); and LLM-specific unlearning approaches, including **ICUL** (Pawelczyk et al., 2024), **SGA** (Jang et al., 2023; Bărbulescu & Triantafillou, 2024), **TAU** (Bărbulescu & Triantafillou, 2024), **CUT** (Li et al., 2024b), **NPO** (Zhang et al., 2024b), and **SOUL-GRADDIFF** (Jia et al., 2024). For ICUL (Pawelczyk et al., 2024), we randomly select one example $(q_i, y_i)$ from $\mathcal{T}_f$ and corrupt the output $y_i$ with randomly selected tokens. Then we concatenate this corrupted sequence with other intact sequences as the in-context demonstrations. For all other baselines, we treat all data related to $\mathcal{T}_f$ as unlearning examples and all data related to $\mathcal{T}_r$ as remaining examples. Everything else remains the same for each baseline.

## 5. Results

**Comparison to general unlearning methods** Our main results in Table 1 show that TOOLDELETE outperforms general unlearning baselines. Compared to RETRAIN, the best-performing baseline, TOOLDELETE-SFT achieves gains of 0.6, 0.3, 8.0, 2.3 absolute points on $\mathcal{T}_T, \mathcal{T}_r, \mathcal{T}_f, \mathcal{T}_G$ respectively. TOOLDELETE-DPO shows even stronger results, outperforming RETRAIN by 1.3, 3.3, 9.8, 1.8 points on the same metrics. We note that GRADASCENT can effectively unlearn $\mathcal{T}_f$, but it negatively impacts its $\mathcal{T}_T$ and $\mathcal{T}_r$ performance. Although RANDLABEL and SALUN outperforms GRADASCENT, they still fall short on $\mathcal{T}_G$ compared to TOOLDELETE.

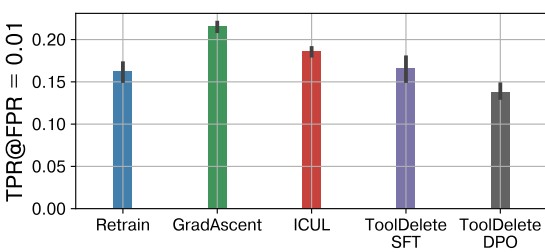

*Figure 2.* Measuring tool unlearning with LiRA-Tool.

*Table 2.* Ablation study of proposed properties on ToolAlpaca. Highlighted are metrics that degrade after removing specific parts of the model.

| | TOOLDELETE-SFT | | | | TOOLDELETE-DPO | | | |
|---|---|---|---|---|---|---|---|---|
| | $\mathcal{T}_{\mathbf{T}}(\uparrow)$ | $\mathcal{T}_{\mathbf{r}}(\uparrow)$ | $\mathcal{T}_{\mathbf{f}}(\downarrow)$ | $\mathcal{T}_{\mathbf{G}}(\uparrow)$ | $\mathcal{T}_{\mathbf{T}}(\uparrow)$ | $\mathcal{T}_{\mathbf{r}}(\uparrow)$ | $\mathcal{T}_{\mathbf{f}}(\downarrow)$ | $\mathcal{T}_{\mathbf{G}}(\uparrow)$ |
| FULL | **57.7** | **72.1** | **30.5** | **23.6** | **58.4** | **73.3** | **28.7** | **23.1** |
| - TKD | 58.1 | 72.4 | 65.3 | 23.3 | 58.6 | 73.2 | 65.9 | 22.7 |
| - TKR | 32.7 | 40.2 | 23.1 | 20.1 | 40.3 | 41.8 | 39.3 | 22.1 |
| - GCR | 58.0 | 72.5 | 31.1 | 17.5 | 55.7 | 72.7 | 33.1 | 14.3 |

**Comparison to LLM-specific unlearning methods** Existing LLM unlearning methods, despite effective in sample-level unlearning, are prone to under-performing in tool unlearning. Both TOOLDELETE-SFT and TOOLDELETE-DPO outperforms ICUL, SGA, and TAU on $\mathcal{T}_T$, $\mathcal{T}_r$, $\mathcal{T}_f$ and $\mathcal{T}_G$. The only exception is ICUL, which outperforms TOOLDELETE-SFT on $\mathcal{T}_r$ by 2.7 absolute points, but is outperformed by TOOLDELETE-DPO on $\mathcal{T}_r$ by 0.3 points. The good performance of ICUL on $\mathcal{T}_r$ is at the cost of failing to unlearn tools in $\mathcal{T}_f$, which is not desired in tool unlearning. In addition, ICUL has limited ability of preserving test set performance, it is outperformed by TOOLDELETE-SFT and TOOLDELETE-DPO by 3.6 and 4.3 respectively. Furthremore, it is particularly limited in deletion capacity, i.e. number of unlearning samples that a method can handle. As $|D_f|$ exceeds 10, the performance of ICUL on $\mathcal{T}_T$ significantly degrades. This is while TOOLDELETE can process much larger deletion requests efficiently.

**SFT vs. DPO** DPO outperforms SFT by 0.7, 3.0, and 1.8 on $\mathcal{T}_T$, $\mathcal{T}_r$, $\mathcal{T}_f$ respectively. On $\mathcal{T}_G$, SFT is slightly better than DPO by 0.5 points. However, DPO takes slightly longer time to train, see Figure 4 in Appendix D. Both optimization methods achieve superior performance over existing approaches.

**Measuring tool unlearning with MIA** Following prior practices (Carlini et al., 2022; Pawelczyk et al., 2024), a lower TPR indicates an unlearned model with better privacy when FPR=0.01. TOOLDELETE-DPO achieves 0.14 TPR, outperforming RETRAIN by 0.01. This advantage is obtained by explicitly prioritizing tool-free responses $f_0(\mathcal{Q})$ over original responses. In addition, TOOLDELETE-SFT achieves comparable performance with RETRAIN, which indicates its effectiveness to protect privacy. Both variants of our method outperforms GRADASCENT and ICUL, the best performing baselines, achieving 0.21 and 0.18 TPR. This indicates that existing sample-level unlearning approaches are not sufficient for unlearning tools, see Figure 2.

**Sequential unlearning** Tool unlearning requests may arrive in sequential mini-batches. We experiment with sequen-

tial unlearning requests by incrementally unlearning 2%, 5%, 10%, and 20% of tools. RETRAIN, ICUL by design cannot process sequential deletion requests. TOOLDELETE can continue training according to the current deletion request, without having to retrain a new model. When 20% of unlearning requests arrive in batches, TOOLDELETE can sequentially unlearn each of them. As Figure 3 and Table 1 show, compared to unlearning 20% at once, the performance does not degrade significantly.

**All properties contribute to effective tool unlearning** Ablation studies in Table 2 show that without Tool Knowledge Removal, performance of TOOLDELETE-SFT and TOOLDELETE-DPO on $\mathcal{T}_f$ degrade by -34.8 and -37.2 absolute points respectively. Such significant performance drop is observed for other model properties as well. Therefore, we conclude all proposed properties are necessary for successful at tool unlearning on $\mathcal{T}_T$, $\mathcal{T}_r$, $\mathcal{T}_f$, and $\mathcal{T}_G$.

**TOOLDELETE functions effectively without access to training data** In certain unlearning settings, access to the original training data might be restricted, e.g., in healthcare settings or in cases where training data is no longer available due to compliance. In these cases, TOOLDELETE can generate pseudo-samples for tools using the "shadow samples" technique developed for LiRA-Tool, see §3.5. Table 4 in Appendix D shows that TOOLDELETE can perform tool unlearning effectively, achieving comparable performances to when full access to the exact training data is available.

**TOOLDELETE is efficient** Efficiency is a critical aspect for unlearning. As Figure 4 illustrates, TOOLDELETE is substantially more efficient than retraining a new model from scratch–saving about 74.8% of training time on average. In addition, this efficiency gain is relatively consistent as the size of $T_f$ increases. TOOLDELETE-SFT is slightly faster than TOOLDELETE-DPO, as the latter requires a negative sample for each of its prompts.

**TOOLDELETE-LoRA is ultra-efficient with good unlearning performance** We experiment if TOOLDELETE can achieve effective tool unlearning through LoRA (Hu et al., 2022), when computing resource is limited. Experi-

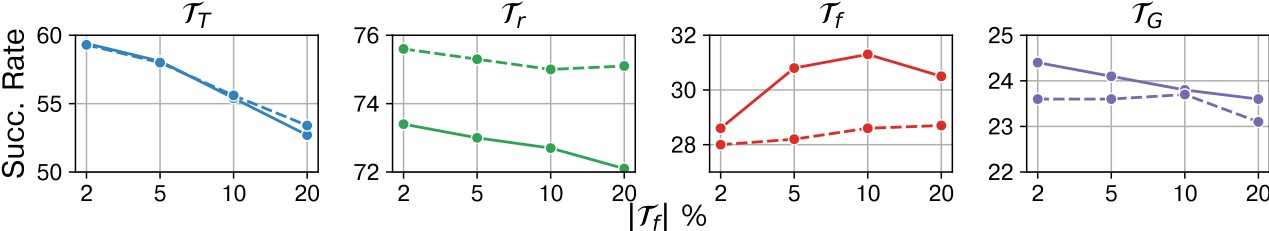

*Figure 3.* Performance of sequential unlearning on ToolAlpaca. We unlearn 2%, 5%, 10%, 20% of tools in a sequential manner.

ments on ToolAlpaca show that TOOLDELETE-LoRA can achieve 97.7%, 99.6%, 84.5%, and 84.3% of the performance of TOOLDELETE with full parameter on $\mathcal{T}_T$, $\mathcal{T}_r$, $\mathcal{T}_f$, $\mathcal{T}_G$ on average across SFT and DPO, see Table 3 in Appendix D. In addition, it reduces save computational cost by 81.1% and decreases the training time by 71.3%.

**TOOLDELETE is flexible in choice of tool-free responses** In (1), we obtain tool knowledge-free responses from the tool-free LLM $f_0$. However, in cases where $f_0$ is unavailable, TOOLDELETE can still function using any knowledge-free LLM to generate tool knowledge-free responses, such as a randomly initialized LLM $f_R$. Table 7 compares the performances between these two implementations. While $\theta_0$ consistently outperforms $\theta_R$, using $\theta_R$ is still effective in achieving tool unlearning.

**Why is TOOLDELETE effective?** We attribute the performance of TOOLDELETE to its three key properties: (a): Tool Knowledge Removal enables targeted tool unlearning without over-forgetting, unlike GRADASCENT and RETRAIN. This is achieved by prioritizing tool knowledge-free responses over tool knowledge-intense responses so that the model forgets tool functionality without excessive degradation. This formulation imposes the right strength of forgetting over specific tools, while existing methods may over- or under-unlearn. (b): Tool Knowledge Retention reinforces the knowledge about remaining tools. In fact, re-exposing the model to the original training data can further strengthen their representation. (c): General Capability Retention, which maintains or even improves model's general capabilities through an efficient and effective task arithmetic operation. Therefore, precise unlearning, retention of relevant knowledge, and overall model stability are the key factors that contribute to the performance of TOOLDELETE.

## 6. Related work

**Unlearning for non-LLM models**: These methods include methods that focus on pruning before unlearning (Jia et al., 2023) or finding salient parameters (Fan et al., 2024) and manipulating gradients (Ullah et al., 2021; Hoang et al., 2024),

adversarial methods (Liu et al., 2023; Setlur et al., 2022; Wei et al., 2023), approximation of inverse Hessian (Zhang et al., 2024a), and data augmentation (Choi et al., 2024). Other works study unlearning under multimodal setting (Cheng & Amiri, 2024a), image-to-image models (Li et al., 2024a), and finding the most challenging unlearning subset within a dataset (Fan et al., 2025b). Recently, a few works started to benchmark MU performances on unlearning fictitious user profiles (Maini et al., 2024), world knowledge (Jin et al., 2024) and a variety of tasks (Cheng & Amiri, 2024b).

**Unlearning for LLMs**: Recently, more attention has been given to LLM unlearning, where gradient ascent is a common technique (Eldan & Russinovich, 2023; Jang et al., 2023). (Yao et al., 2024) evaluate several traditional unlearning methods on LLMs. KGA (Wang et al., 2023) formulates unlearning as achieving knowledge gap between training data and test data similar to that of training data and deleted data. Yao et al. (2023) proposed to predict if the LLM output is grammatically correct on deleted samples, such that the knowledge is not over unlearned. Other methods include second-order-optimization (Jia et al., 2024), performing DPO with no positive examples (Zhang et al., 2024b), and reinforcement learning with a negative reward model (Kassem et al., 2023). Unlearning from logits difference (Ji et al., 2024) first builds an assisted LLM which memorizes data to be deleted and forgets the retained data, which is later used to derive the unlearned LLM by deviating from the assisted LLM in logits.

**Tool-Augmented LLMs**: Tool augmented language models (TAML) (Parisi et al., 2022) used self-play to boost LLMs' performance on math and reasoning tasks. In addition, Toolformer (Schick et al., 2023) showed that LLMs can teach themselves how to use APIs. More recent efforts have been devoted to building benchmarks to train and evaluate the tool-using ability of LLMs. These include agent-based data generation (Tang et al., 2023; Li et al., 2023), bootstrapping training data with various seed examples (Patil et al., 2023), modifying existing datasets (Basu et al., 2024), and dataset development with powerfull LLMs such as GPT-4 (Qin et al., 2024).

# 7. Conclusion

We introduce Tool Unlearning–a novel machine unlearning task with the goal of unlearning previously learned tools from tool-augmented LLMs. We develop the first tool unlearning approach, TOOLDELETE, that implements three key properties: *tool knowledge deletion*, *tool knowledge retention*, *general capability retention*. In addition, we introduce LiRA-Tool, the first membership inference attack (MIA) method for evaluating tool unlearning. LiRA-Tool largely addresses the limitations of sample-based MIAs for tool unlearning. Extensive experiments on several diverse datasets and LLMs show that TOOLDELETE is an efficient, flexible, and effective tool unlearning method that supports sequential unlearning, maintains strong performance across all key properties, and operates without requiring full access to training data. It outperforms existing methods by removing tool knowledge without over-forgetting (as shown in ablation studies), achieving 74.8% faster training times compared to retraining, and delivering highly effective tool unlearning even in resource-constrained settings with TOOLDELETE-LoRA (which reduces compute costs by 81.1% and training time by 71.3%). In future, we will investigate tool unlearning in continually updated LLMs to address continuous unlearning challenges. In addition, we will develop adversarial training techniques and robustness evaluation frameworks to prevent unintended tool re-learning or model exploitation (Fan et al., 2025a), and conduct loss landscape analysis of tool unlearning (Cheng & Amiri, 2025)

**Limitations** We did not conduct experiments using closed-source LLMs or API-based LLMs. In addition, this work did not investigate the impact of varying model scales due to the limited publicly-available tool-augmented LLMs. Our experiments were conducted on the 7B scale and the scalability of the proposed tool unlearning approach across models of different sizes and scales is an open question for future investigation. Moreover, evaluation of the efficacy of tool unlearning can be extended to broader conditions, such as under adversarial conditions (Łucki et al., 2025).

# Impact Statement

Our work investigates machine unlearning in the context of tool-augmented Large Language Models (LLMs), where we focus on the risks that arise from integrating external tools and the crucial need for unlearning tool-usage capabilities for specific tools to ensure compliance with privacy regulations such as the Right to be Forgotten (RTBF). This necessitates the ability to delete sensitive, regulated, or outdated knowledge related to specific tools. Tool unlearning will enable us to identify potential threats to model security, e.g. unauthorized tool usage, adversarial exploitation, and privacy violations. Our research highlights the importance of addressing these challenges.

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

# A. Practical Use Cases of Tool Unlearning

We provide several examples in which tool unlearning is essential:

**Case 1: De-memorize Privacy-Concerned Tools** Imagine a tool-augmented LLM that is deployed in a healthcare system and trained to use APIs for handling and processing patient data, such as accessing medical records or generating anonymized reports. Suppose one of the APIs that was initially compliant is later flagged as insecure due to a vulnerability that could expose patient data. This violates regulations like HIPAA or GDPR. In this case, ToolDelete is essential as it can update the tool-augmented LLM's parameters to unlearn how to invoke the insecure API. This removes any capability embedded in the LLM's parametric knowledge and prevents adversarial or accidental usage of the vulnerable API.

**Case 2: Forget Harmful / Biased Tools** Consider a tool-augmented LLM that can use a Safe For Work diffusion model as a tool to generate images based on user instructions. If the user prompts can fool the model to generate Not Safe For Work (NSFW), harmful, or biased images, this tool should be unlearned from the LLM. Note that even if we augment the LLM with a new and safe version of the diffusion model without unlearning the previous version, the LLM would still be able to call the previous version, which can lead to generating Not Safe For Work, harmful, or biased images. Therefore, we should explicitly erase the ability of using the previous version of the diffusion model from the LLM.

**Case 3: Unlearn Deprecated Tools** Tool unlearning is also essential when a tool has a major update, where the function names and input parameters have changed, e.g. the major update of the Python transformers package from v2 to v4. Without unlearning v2, the tool-augmented LLM may generate erroneous code and bring difficulty for debugging, since many functions have been renamed and removed. Therefore, as the underlying tools evolve, the tool-augmented LLM should be updated through unlearning of the previous versions and augmenting the new ones.

# B. Baselines

As there are no prior works on tool unlearning, we adapt the following unlearning methods to tool unlearning setting. Four general unlearning approaches.

- **GRADASCENT** (Golatkar et al., 2020; Yao et al., 2024) runs gradient ascent on $\mathcal{T}_f$ with the associated query-reponse samples $(\mathcal{Q}_f, \mathcal{Y}_f)$.

- **RANDLABEL** (Graves et al., 2021) fine-tunes on $\mathcal{T}_r$ and $\mathcal{T}_f$ with corrupted labels.

- **SALUN** (Fan et al., 2024) performs RANDLABEL on unlearning-related parameters discovered by saliency map.

- **ICUL** (Pawelczyk et al., 2024) uses $\mathcal{T}_f$ with corrupted label as in-context demonstrations.

- **SGA** (Jang et al., 2023; Bărbulescu & Triantafillou, 2024), which performs gradient ascent on $\mathcal{T}_f$ whose memorization probability exceeds a pre-defined threshold.

- **TAU** (Bărbulescu & Triantafillou, 2024), which performs task arithmetic on SGA.

- **CUT** (Li et al., 2024b), which controls model activations to be similar to the absence of knowledge on forget set.

- **NPO** (Zhang et al., 2024b) uses DPO with only a losing response (i.e. no winning response).

- **SOUL-GradDiff** (Jia et al., 2024) uses second-order information in optimization. It adapts the Sophia optimizer (Liu et al., 2024) for LLM unlearning. We adopt the SOUL + GradDiff (Maini et al., 2024) implementation in the original paper.

# C. Implementation details

We use a learning rate of $10^{-5}$ across all experiments. All experiments are conducted on 8 NVIDIA A100 GPUs.

For the original models in tool unlearning, we use the `TangQiaoYu/ToolAlpaca-7B`, `ToolBench/ToolLLaMA-2-7b-v2`, `gorilla-llm/gorilla-openfunctions-v0` checkpoints that are publically available on Huggingface.

*Table 3.* Full parameters vs. LoRA in tool unlearning performances when deleting 20% of tools on ToolAlpaca. *Original* denotes the tool-augmented LLM prior unlearning and is provided *for reference only*.

|  | $\mathcal{T}_T(\uparrow)$ | $\mathcal{T}_r(\downarrow)$ | $\mathcal{T}_f(\uparrow)$ | $\mathcal{T}_G(\uparrow)$ |
|---|---|---|---|---|
| ORIGINAL (PRIOR UN.) | 60.0 | 73.1 | 75.7 | 24.1 |
| FULL PARAM | 52.7 | 72.1 | 30.5 | 23.6 |
| LORA | 51.5 | 71.8 | 36.1 | 19.9 |

*Table 4.* Performance comparison between with and without having access to the exact training samples.

| METHOD | $\mathcal{T}_t(\uparrow)$ | $\mathcal{T}_r(\uparrow)$ | $\mathcal{T}_f(\downarrow)$ | $\mathcal{T}_G(\uparrow)$ |
|---|---|---|---|---|
| *W/ access to training samples* | | | | |
| TOOLDELETE-SFT | 52.7 | 72.1 | 30.5 | 23.6 |
| TOOLDELETE-DPO | 53.4 | 75.1 | 28.7 | 23.1 |
| *W/o access to training samples* | | | | |
| TOOLDELETE-SFT | 52.0 | 72.5 | 30.1 | 22.8 |
| TOOLDELETE-DPO | 52.9 | 76.0 | 28.0 | 22.5 |

# D. Additional results

We present the results of LoRA tool unlearning, sequential tool unlearning, time comparison and results on ToolLLaMA and Gorilla in Table 3–6.

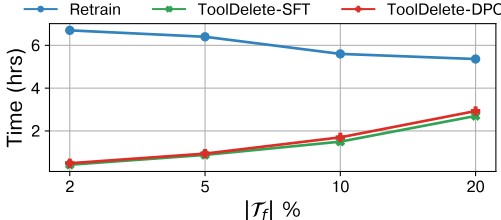

*Figure 4.* Training time of TOOLDELETE, which saves 74.8% of time on average.

*Table 5.* Tool unlearning performances when deleting 20% of tools on ToolLLaMA. Best and second best performances are **bold** and underlined respectively. Original denotes the tool-augmented LLM prior unlearning and is provided for reference only .

| Method | $\mathcal{T}_T(\uparrow)$ | $\mathcal{T}_r(\uparrow)$ | $\mathcal{T}_f(\downarrow)$ | General Capability $\mathcal{T}_G(\uparrow)$ | | | | |
| --- | --- | --- | --- | --- | --- | --- | --- | --- |
| | | | | STEM | Reason | Ins-Follow | Fact | Avg. |
| Original (Prior Un.) | 64.0 | 75.6 | 76.0 | 25.3 | 36.8 | 17.3 | 15.0 | 23.6 |
| *General Unlearning Methods* | | | | | | | | |
| RETRAIN | 62.2 | 72.1 | 42.3 | 25.1 | 33.7 | 14.6 | 13.8 | 21.8 |
| GRADASCENT | 42.5 | 56.3 | 51.8 | 14.9 | 26.4 | 11.2 | 8.6 | 15.3 |
| RANDLABEL | 59.3 | 73.5 | 40.7 | 23.4 | 30.6 | 13.3 | 12.7 | 20.0 |
| SALUN | 58.7 | 73.6 | 39.9 | 22.7 | 30.8 | 13.6 | 12.0 | 19.8 |
| *LLM-Specific Unlearning Methods* | | | | | | | | |
| ICUL | 46.2 | 68.2 | 57.2 | 15.1 | 18.8 | 7.1 | 9.4 | 12.6 |
| SGA | 44.7 | 59.6 | 49.4 | 16.3 | 20.4 | 12.8 | 9.7 | 14.8 |
| TAU | 44.5 | 56.3 | 50.2 | 21.6 | 28.0 | 15.3 | 13.5 | 19.6 |
| CUT | 52.4 | 59.5 | 44.2 | 20.7 | 24.1 | 13.7 | 12.8 | 17.8 |
| NPO | 58.3 | 66.3 | 40.2 | 23.0 | 31.7 | 15.4 | 11.9 | 20.5 |
| SOUL-GradDiff | 62.2 | 70.4 | 40.7 | 24.2 | 28.6 | 14.7 | 12.2 | 19.9 |
| TOOLDELETE-SFT | 62.8 | 72.8 | 39.5 | 24.6 | 33.4 | 15.8 | 13.7 | **21.9** |
| TOOLDELETE-DPO | **63.2** | **73.6** | **38.7** | 24.3 | 32.9 | 16.0 | 13.8 | 21.8 |

*Table 6.* Tool unlearning performances when deleting 20% of tools on ToolLLaMA. Best and second best performances are **bold** and underlined respectively. Original denotes the tool-augmented LLM prior unlearning and is provided for reference only .

| Method | $\mathcal{T}_T(\uparrow)$ | $\mathcal{T}_r(\uparrow)$ | $\mathcal{T}_f(\downarrow)$ | General Capability $\mathcal{T}_G(\uparrow)$ | | | | |
| --- | --- | --- | --- | --- | --- | --- | --- | --- |
| | | | | STEM | Reason | Ins-Follow | Fact | Avg. |
| Original (Prior Un.) | 64.0 | 75.6 | 76.0 | 25.3 | 36.8 | 17.3 | 15.0 | 23.6 |
| *General Unlearning Methods* | | | | | | | | |
| RETRAIN | 62.2 | 72.1 | 42.3 | 25.1 | 33.7 | 14.6 | 13.8 | 21.8 |
| GRADASCENT | 42.5 | 56.3 | 51.8 | 14.9 | 26.4 | 11.2 | 8.6 | 15.3 |
| RANDLABEL | 59.3 | 73.5 | 40.7 | 23.4 | 30.6 | 13.3 | 12.7 | 20.0 |
| SALUN | 58.7 | 73.6 | 39.9 | 22.7 | 30.8 | 13.6 | 12.0 | 19.8 |
| *LLM-Specific Unlearning Methods* | | | | | | | | |
| ICUL | 46.2 | 68.2 | 57.2 | 15.1 | 18.8 | 7.1 | 9.4 | 12.6 |
| SGA | 44.7 | 59.6 | 49.4 | 16.3 | 20.4 | 12.8 | 9.7 | 14.8 |
| TAU | 44.5 | 56.3 | 50.2 | 21.6 | 28.0 | 15.3 | 13.5 | 19.6 |
| CUT | 52.4 | 59.5 | 44.2 | 20.7 | 24.1 | 13.7 | 12.8 | 17.8 |
| NPO | 58.3 | 66.3 | 40.2 | 23.0 | 31.7 | 15.4 | 11.9 | 20.5 |
| SOUL-GradDiff | 62.2 | 70.4 | 40.7 | 24.2 | 28.6 | 14.7 | 12.2 | 19.9 |
| TOOLDELETE-SFT | 62.8 | 72.8 | 39.5 | 24.6 | 33.4 | 15.8 | 13.7 | **21.9** |
| TOOLDELETE-DPO | **63.2** | **73.6** | **38.7** | 24.3 | 32.9 | 16.0 | 13.8 | 21.8 |

*Table 7.* Performance comparison between using pre-trained LLM $f_0$ and randomly initialized LLM $f_R$.

| METHOD | $\mathcal{T}_t(\uparrow)$ | $\mathcal{T}_r(\uparrow)$ | $\mathcal{T}_f(\downarrow)$ | $\mathcal{T}_G(\uparrow)$ |
|---|---|---|---|---|
| *Pre-trained LLM weights $f_0$* | | | | |
| TOOLDELETE-SFT | 52.7 | 72.1 | 30.5 | 23.6 |
| TOOLDELETE-DPO | 53.4 | 75.1 | 28.7 | 23.1 |
| *Randomly initialized LLM $f_R$* | | | | |
| TOOLDELETE-SFT | 50.9 | 71.3 | 29.8 | 22.7 |
| TOOLDELETE-DPO | 52.6 | 73.4 | 27.5 | 22.4 |

# E. Sampling of Shadow Samples for LiRA-Tool

We use the following prompt to prompt GPT-4 to synthesize diverse shadow samples for evaluation with LiRA-Tool.

> You are now a synthetic data generator. Generate query-response pairs to evaluate an LLM's ability of using an API.
> How to generate "query": Based on the API and documentation shown below, think of a user query that needs to be answered by calling the API.
> How to generate "response": Write down the correct API call with correct arguments.
> The in-context examples below demonstrate what you need to generate. Please be as diverse and creative as possible in phrasing and style. But do not hallucinate.
> ## In-context Examples #### Tool and Documentation Name: StableDiffusionPipeline.from_pretrained()
> #### Query I want to see some cats dancing in celebration!
> #### Response API call: StableDiffusionPipelin e.from_pretrained("stabilityai/stable-diffusion-2-1")
> Now, for the following API, generate a query-response pair.
> #### Tool and Documentation api_name()
> #### Query
> #### Response

