# OpenReview forum: "Tool Unlearning for Tool-Augmented LLMs"
_ICML.cc/2025/Conference — ICML 2025 poster_

### Official Review · Reviewer_qa8k · 2025-03-01

**Overall Recommendation:** 2

**Summary:**

The authors introduce tool unlearning, a novel task in the machine unlearning domain, motivated by the need to remove learned tools from tool-augmented LLMs due to security, privacy, or obsolescence concerns. Unlike traditional unlearning, this task presents unique challenges, including knowledge removal beyond individual sample forgetting, the high computational cost of optimizing LLMs, and the lack of principled evaluation metrics. To address these challenges, the authors propose ToolDelete, the first dedicated approach for tool unlearning, which incorporates three key properties for effective unlearning. Additionally, they introduce a new membership inference attack (MIA) model for evaluation. Experimental results on three tool-learning datasets demonstrate that ToolDelete successfully unlearns both randomly selected and category-specific tools while preserving the model’s general knowledge and maintaining performance on non-deleted tools.

**Claims And Evidence:**

I think the authors overclaim their contributions. I discuss them in Questions For Authors.

**Essential References Not Discussed:**

N/A

**Experimental Designs Or Analyses:**

All of them. I discuss them in Questions For Authors.

**Methods And Evaluation Criteria:**

The author did not propose a new method but applied SFT/DPO to the Tool benchmark.

**Other Comments Or Suggestions:**

I discuss them in Questions For Authors.

**Other Strengths And Weaknesses:**

I discuss them in Questions For Authors.

**Questions For Authors:**

1. I believe that the standard unlearning task is not strictly limited to sample-level unlearning. For example, WMDP focuses on forgetting sensitive concepts such as biology, while MUSE Books aims to forget concepts related to Harry Potter. Based on the authors’ definition of Tool Unlearning, it seems more aligned with in-context unlearning [1]. Therefore, I feel the authors may be overstating their novelty.

[1] Pawelczyk, Martin, Seth Neel, and Himabindu Lakkaraju. "In-context unlearning: Language models as few-shot unlearners." arXiv preprint arXiv:2310.07579 (2023).

2. The contribution of LiRA-Tool also seems overstated. The method essentially applies LiRA to the current setting without introducing significant innovation. Additionally, the paper does not provide any analysis or experimental evidence explaining why traditional MIA (Membership Inference Attacks) perform poorly in the Tool Unlearning setting.

3. The authors do not propose any new unlearning method. Instead, they simply apply SFT and DPO to existing benchmarks (ToolAlpaca, ToolBench, etc.), without introducing a novel approach to unlearning.

4. Hyperparameter impact: Compared to standard baselines, the proposed method introduces several additional hyperparameters from TOOLDELETE and SFT/DPO. While TOOLDELETE clearly introduces even more hyperparameters, the paper does not analyze their impact on the final results, which is a crucial omission.

**Relation To Broader Scientific Literature:**

I discuss them in Questions For Authors.

**Theoretical Claims:**

No theoretical Claims.

---

> ### Author Rebuttal · Authors · 2025-04-01
>
> **Re W1: I believe that the standard unlearning task is not strictly limited to sample-level unlearning. For example, WMDP focuses on forgetting sensitive concepts such as biology, while MUSE Books aims to forget concepts related to Harry Potter. Based on the authors’ definition of Tool Unlearning, it seems more aligned with in-context unlearning [1]. Therefore, I feel the authors may be overstating their novelty.**
>
> Thank you for raising this important point. Tool unlearning differs from both concept-level and in-context unlearning in several important ways:
>
> (a): parametric capability removal vs. concept forgetting: concept-level unlearning  in WMDP and MUSE Book mainly focus on “semantic or factual concepts” embedded in the model’s knowledge. In contrast, tool unlearning focuses on “functional capabilities,” i.e. the ability to perform a task using a specific tool or API, which is action-oriented as opposed to semantic-oriented modifications in concept-level unlearning.  We acknowledge the similarity and will make the description clear in the revised version.
>
> (b): In-context unlearning operates at the “prompt level” without modifying model parameters. As we show in our experiments (e.g. ICUL, Table 1 and Line 316--329), this form of unlearning is insufficient for tool unlearning, since the parametric knowledge of the tool remains and can be invoked with adversarial or indirect prompts.
>
> We will make these distinctions clear in the revised version.
>
>
>
> **Re W2: The contribution of LiRA-Tool also seems overstated. The method essentially applies LiRA to the current setting without introducing significant innovation. Additionally, the paper does not provide any analysis or experimental evidence explaining why traditional MIA (Membership Inference Attacks) perform poorly in the Tool Unlearning setting.**
>
> We respectfully argue that the adaptation of LiRA to tool unlearning is both necessary and non-trivial. Traditional LiRA targets sample-level membership, which is insufficient for evaluating whether a model has truly forgotten a tool-using capability, which requires evaluating broader functional capability. LiRA-Tool introduces two key innovations: (1): shifting the evaluation from sample-level membership to skill-level forgetting, and (2): using shadow samples to robustly probe tool knowledge beyond the training data.
>
> Recent works have found that LLM unlearning is susceptible to re-learning attack (Hu et al., ICLR 2025) or jailbreak attack (Lynch et al., 2024). LiRA-Tool stress-tests the unlearned model, ensuring that a specific tool has truly been forgotten, providing better demonstration of successful tool unlearning, instead of only sample de-memorization.
>
>
> **Re W3: The authors do not propose any new unlearning method. Instead, they simply apply SFT and DPO to existing benchmarks (ToolAlpaca, ToolBench, etc.), without introducing a novel approach to unlearning.**
>
> We would like to clarify that the ToolDelete framework, its unlearning objective formulation—Equations (1-4)—and its approach to general capability retention via task arithmetic are novel and specifically designed for tool unlearning. We believe these contributions are substantial and fill an existing gap in current unlearning research. In addition, we clarify that ToolDelete uses SFT and DPO simply as training methods.
>
>
> **Re W4: Hyperparameter impact: Compared to standard baselines, the proposed method introduces several additional hyperparameters from TOOLDELETE and SFT/DPO. While TOOLDELETE clearly introduces even more hyperparameters, the paper does not analyze their impact on the final results, which is a crucial omission.**
>
> ToolDelete introduces a few additional parameters and we have analyzed them throughout the main paper and Appendix. Specifically:
> - $\alpha$ in Eq (6): we tuned $\alpha$ on a held-out validation set.
> - Training method (SFT vs. DPO): The results of both variants were shown in Table 1.
> - Access to training data: In Appendix D, Table 4, we reported performance comparison with and without having access to the exact training samples.
> - PEFT vs. full-parameter tuning: In Appendix D, Table 3, we compared LoRA with full parameter tuning.
> - Choice of tool-free response in Eq (1): In Appendix D, Table 5, we evaluated the impact of using different sources for tool-free responses (e.g. $f_0$ vs. $f_R$).
>
> We will make sure these references are explicitly explained in the final version of the paper, and provide a brief summary table of hyperparameters and their effects for clarity. We appreciate the reviewer highlighting this point.

---

### Official Review · Reviewer_tu3r · 2025-03-08

**Overall Recommendation:** 3

**Summary:**

This paper emphasizes an emergent problem that LLMs need to unlearn tools that have potential security concerns, and therefore proposes the ToolDelete method to remove the knowledge of using specified tools, as well as introducing an adapted membership inference attack (MIA) method to evaluate the tool unlearning progress. The experiments show superior model performance in forgetting undesired tools, maintaining knowledge about other tools and general arithmetic abilities.

**Claims And Evidence:**

1. The proposed ToolDelete method can effectively remove knowledge about unwanted tools and maintain knowledge about other tools. Results in Table 1 show decreased and improved results on the undesired and the remaining tools. However, Table 1 reports task-solving accuracy, where a low accuracy could be: (i) the model not using the tool (which is good), or (ii) the model still heavily uses the tool but incorrectly. It is unclear if (ii) should be considered good; and if not, the reported results mix (ii) into the main aspect of interest (i.e., (i)).
2. Justification for effectiveness of the LiRA-Tool evaluation: While the method design involves some added modules as introduced in Section 3.5, the results in Figure 2 may not be sufficient to justify the effectiveness of the LiRA-Tool. As Table 1 is not convincing enough to show ToolDelete is a better tool-unlearning method, and the accuracy reporting concern (as in the point above), LiRA-Tool showing similar results may not justify its effectiveness. That being said, comparing other baseline MIA methods could help strengthen the point that LiRA-Tool is valuable.

**Essential References Not Discussed:**

The related work section discuss unlearning approaches rather comprehensively. However, regarding the MIA-based LiRA-Tool for evaluation, the related work does not discuss too much related work to it, and may benefit from more systematic comparisons to these evaluation methods.

**Experimental Designs Or Analyses:**

The experimental design looks reasonable overall. However, a few points I found a bit confusing:
1. The “Datasets & Tool-Augmented LLMs” in section 4 mentions three datasets (ToolAlpaca, ToolBench, API-Bench) for evaluation, however all following experiments are only performed on ToolAlpaca.
2. The “Setup & Evaluation” paragraph afterward says “with 2-20% tools randomly selected”, but Table 1 caption says “deleting 20%” which describes different setups, please align the setup description and accurately describe the experiments.
3. If a fixed proportion (e.g., 20%) of tools is selected for deletion (as in point 2), two important experiments to do is (1) fixing the 20%, how do differently selected tools affect the final evaluation result reported in Table 1 and Figure 2? And (2) how would the choice of tool deletion proportion (e.g., from 20% to 10%, 50%, etc.) affect the experiment results?

**Methods And Evaluation Criteria:**

1. The set of evaluation dimensions are comprehensive and reasonable to study the tool-unlearning problem, including measuring performance on to-forget and remaining tools. Also, evaluating models on general problem abilities (STEM, reason, instruction-following, and facts) is important. A minor point: in Section 3.3, the authors proposes to test general capabilities via Arithmetic tasks, the paragraph “Why Task Arithmetic?” does not seem convincing enough, particularly why arithmetic is core LLM ability and how crucial it is to tool-using. Nonetheless, the list of tasks used in experiments are not necessarily arithmetic tasks (?) but seem reasonable and comprehensive to me. Consider changing the overall description of tasks in Section 3.3 could be an easier fix to this confusion.
2. The proposed LiRA-Tool is a novel and seemingly more targeted evaluation tool for the tool-unlearning problem. However, it is unclear to me: (1) what additional information this MIA-based method tells beyond default accuracy measures, and (2) how much more effective LiRA-tool is compared to standard MIA or similar approaches.

**Other Comments Or Suggestions:**

N/A

**Other Strengths And Weaknesses:**

Strength: The target task – tool unlearning – could be an emerging yet unexplored area, for which it is interesting to develop methods and evaluation procedures.

**Questions For Authors:**

For the tool benchmarks listed for evaluation in Section 4, at least on ToolBench, some tools are no longer executable/available, thus making it difficult to reproduce their original results or conduct fair comparisons. I wonder if the authors have encountered similar issues on ToolBench or the other datasets.

**Relation To Broader Scientific Literature:**

The key contribution of the paper is providing an adapted unlearning method for LLM-used tools. The paper is therefore largely related to knowledge/data uncleaning for LLMs.

**Theoretical Claims:**

The work represents the three key properties of the proposed ToolDelete method, as well as the model training details, in symbolic expressions, which all read reasonable.

---

> ### Author Rebuttal · Authors · 2025-04-01
>
> **Re W1: Table 1 reports task-solving accuracy, where a low accuracy could be: (i) the model not using the tool (which is good), or (ii) the model still heavily uses the tool but incorrectly. It is unclear if (ii) should be considered good; and if not, the reported results mix (ii) into the main aspect of interest (i.e., (i)).**
>
>
> We thank the reviewer for their comment. We agree that distinguishing between (i) and (ii) is important. In the context of tool unlearning, incorrectly using a tool, case (ii), indicates the model no longer has the tool-using capability, which is the goal of tool unlearning. For example, if the model generates incorrect arguments for an API, the API call will fail, leading to an errorful or unusable response, which reflects the loss of practical tool-using ability. However, we fully agree with the reviewer that separating cases (i) and (ii) can provide deeper insights. To do so, we conducted additional evaluation by measuring if the unlearned tool is called by the unlearned model. A lower score means that the unlearned model does not use the unlearned tool post-unlearning.
>
> We can see that after applying ToolDelete, the unlearned model seldom uses the forget tools, outperforming existing unlearning methods, indicating a successful tool unlearning.
>
> | Method           | Case (ii) (%, $\downarrow$) |
> |------------------|---------------------------------------------------------------------------------------|
> | Retrain          | 11.3     |
> | GradAscent       | 27.1|
> | RandLabel        | 27.5 |
> | SalUn            | 25.6     |
> | ICUL             | 72.4     |
> | SGA              | 25.9     |
> | TAU              | 26.4      |
> | ToolDelete - SFT | 12.0 |
> | ToolDelete - DPO | **10.8** |
>
>
>
>
>
> **Re W2: Justification for effectiveness of LiRA-Tool: While the method design involves some added modules as introduced in Section 3.5, the results in Figure 2 may not be sufficient to justify the effectiveness of the LiRA-Tool.  It is unclear to me: (1) what additional information this MIA-based method tells beyond default accuracy measures   (2) how much more effective LiRA-tool is compared to standard MIA or similar approaches.**
>
> The innovation of LiRA-Tool is in focusing on “unlearning functional capability”, which is beyond the samples in the forget set. This is done through evaluating the unlearned model more diverse prompts. As discussed in $\S$3.5 and Figure 2, models can show deceptive surface-level behavior—e.g., low accuracy on a forget set—without truly forgetting the tool, such as suppressing tool knowledge for specific prompts. In fact, recent works have found that LLM unlearning is susceptible to re-learning attacks (Hu et al., ICLR 2025) or jailbreak attacks (Lynch et al., 2024). LiRA-Tool stress-tests the model using broader, large-scale, and more varied samples to provide a more robust estimate of whether the tool-using ability has been removed.
>
>
> **Re W3: three datasets in Section 4, but only results on ToolAlpaca.**
>
> We acknowledge that most of the detailed experiments in the main paper focus on ToolAlpaca as the dataset for case study, space constrained, and the high computational cost of experiments. We also conducted several experiments on ToolBench and API-Bench, see Table 6 and Table 7 in Appendix.
>
>
> **Re W4: unlearn 2-20% tools randomly, but Table 1 caption says “deleting 20%” which describes different setups.**
>
> The 2-20% unlearned set is used in sequential unlearning, and the corresponding results are reported in Appendix D, Figure 3. We will clarify this point in the revised version.
>
>
> **Re W5: If a fixed proportion (e.g., 20%), two important experiments to do is (1) fixing the 20%, how do differently selected tools affect the final evaluation result reported in Table 1 and Figure 2? And (2) how would the choice of tool deletion proportion (e.g., from 20% to 10%, 50%, etc.) affect the experiment results?**
>
> We thank the reviewer for their comment. We have partially addressed both reviewer questions in the current version and plan to expand on them further in future work:
>
> (a): Tool selection at fixed proportion: We partially addressed this via class-wise unlearning experiments, where the entire tool categories are unlearned. The results showed how different selection of tool categories (as opposed to random sampling) affect outcomes.
>
>  (b): Varying deletion proportions: Our sequential learning experiments (Appendix D, Figure 3) show the effect of varying deletion proportions. In the final version, we will include results comparing multiple random subsets to analyze performance variability.

---

### Official Review · Reviewer_wfx6 · 2025-03-14

**Overall Recommendation:** 3

**Summary:**

The paper innovatively introduces and conceptualizes tool unlearning for tool-augmented LLMs. The authors propose a novel tool unlearning method called TOOLDELETE with two variants, which satisfies three key properties: tool knowledge removal, tool knowledge retention, and general capability retain. They further introduce a new membership inference attack for tool unlearning. Extensive experiments on multiple tool learning datasets and tool-augmented LLMs show that TOOLDELETE effectively unlearns randomly selected tools while preserving the LLM’s knowledge on non-deleted tools and maintaining performance on general tasks.

**Claims And Evidence:**

Yes.

**Essential References Not Discussed:**

In the latter part of the section, **Unlearning for non-LLM models**, the works study unlearning under multimodal setting (Cheng & Amiri, 2025) togther with the following should be in the section of **Unlearning for LLMs**.

**Experimental Designs Or Analyses:**

The reviewer has checked all of the experimental designs. The reviewer is confused that the performance results do not include that of TOODELETE without variants.

**Methods And Evaluation Criteria:**

Yes.

**Other Comments Or Suggestions:**

1. It would be better to include the comparison results of training time in the main context.

2. Missing results in the last paragraph of the Introduction part on Line 078: "... ***by + in*** accuracy on forget tools ..."

3. Lack of explanation for $\theta_0-\theta_R$ in the caption of Figure 1.

4. Lack of explantion for $P_U(\cdot)$ and $P_{T_r}(\cdot)$ in formulas (7) and (8).

5. Grammar mistake on Line 390: "Why is TOOLDELETE **effectiveness**"

**Other Strengths And Weaknesses:**

Strengths:

1. The idea is novel. The paper is the first to investigate the unlearning problem of previously learned tools from tool-augmented LLMs. The study of tool unlearning is meaningful.
2. The paper is overall well-structured.

Weaknesses:

The section **Training Details** is quite confusing. The authors do not explain how to train TOOLDELETE using RLHF and quantization. There is no formula description for the training process of the given two variants of TOOLDELETE.

**Questions For Authors:**

What is function $g$ specifically? Is the output of $g$ nonnegative? Why isn't formula (1) written in a form similar to formulas (2) and (3), or just $\mathbb E_{t_i\in \mathcal T_f}[g(f',t_i)]=\epsilon$, where $\epsilon$ is an infinitesimal constant? How is the hyperparameter $\alpha$ selected?

**Relation To Broader Scientific Literature:**

This paper presents work whose goal is to advance the field of machine unlearning, especially the unlearning for tool-augmented LLMs, which is specifically oriented to improve the trustworthiness of LLMs with external tools.

**Theoretical Claims:**

The reviewer has checked the correctness of the theoretical claims.

---

> ### Author Rebuttal · Authors · 2025-04-01
>
> **Re W1: The authors do not explain how to train TOOLDELETE using RLHF and quantization. There is no formula description for the training process of the given two variants of TOOLDELETE.**
>
> Our primary focus in this paper has been on proposing the ToolDelete framework and full implementation and analysis of two variants of ToolDelete: SFT and DPO. As mentioned in Lines 185—193, ToolDelete is also compatible with other training paradigms and optimization techniques, including RLHF and quantization. We acknowledge that the description of RLHF and quantization could have been clearer. To clarify:
>
> **RLHF**: share the same training data as DPO. The only difference is that RLHF requires training a reward model, whereas DPO does not. To illustrate, we describe one possible implementation for RLHF:
> - Collect prompt-response pairs as in DPO.
>   - For each unlearned tool $t_i \in \mathcal{T}_f$, designate the tool-free response as the “winning” response and the tool-knowledge response as the “losing” response.
>   - For each retain tool $t_j \in \mathcal{T}_r$, do the opposite: designate the tool-knowledge response as the “winning” response and the tool-free response as the “losing” response.
> - Train a reward model on these preferences.
> - Train the LLM (policy network) to maximize the reward model using PPO, following the standard RLHF procedure (Stiennon et al., NeurIPS 2020, Ouyang et al., 2022).
>
> **Quantization**:  is not a training technique but a model compression and efficiency technique that can be applied with ToolDelete after training with SFT, DPO, or RLHF. Therefore, it does not affect the core unlearning mechanism.
>
> We will make these points clear in the revised version and defer the RLHF variant and quantization experiments to future work.
>
>
> **Re Q1: What is function $g$ specifically? Is the output of $g$ nonnegative? Why isn't formula (1) written in a form similar to formulas (2) and (3), or just $\mathbb{E}_{t_i \in \mathcal{T}_f} [ g(f', t_i) ] = \epsilon$, where $\epsilon$ is an infinitesimal constant?**
>
> The function $g(f, t)$ quantifies the model’s knowledge on tool $t$, such as prompting and evaluating $f$ for ability to use $t$, and is nonnegative. We use Equation (1),  $\mathbb{E}_{t_i \in \mathcal{T}_f} [ g(f_0, t_i) - g(f', t_i) ] \geq 0$, to explicitly compare the unlearned model $f’$ to the vanilla model $f_0$ to make sure $f’$ retains no more knowledge that $f_0$. As we mentioned in Lines 94—96, $f_0$ may already have some tool-using capabilities prior to explicit unlearning (such as basic arithmetic operations). Using our formulation in Equation (1) would give more flexibility in how much tool knowledge we want to unlearn, as we described in Line 135—142. In fact, only using $g(f', t_i) = \epsilon$ would ignore this variation and risk being either too strict or too lenient depending on the tool.
>
> **Re Q2: How is the hyperparameter $\alpha$ selected?**
>
> We select $\alpha$ (which controls the effect of task arithmetic ) through grid search on a held-out validation set.
>
>
>
> **Re Suggestion 4: Lack of explanation for $P_U(\cdot)$ and $P_{T_r}(\cdot)$ in formulas (7) and (8).**
>
> We clarify that $P_U(\cdot)$ indicates the distribution of unlearned tools $T_f$ under the unlearned model $f'$, while $P_{T_r}(\cdot)$ denotes the distribution of the retain tools $T_r$ under the retained model $f$. We thank the reviewer for their suggestion. We will incorporate these definitions explicitly in the paper.
>
>
> **Re: Suggestions.**
>
> We thank the reviewer for their  helpful suggestions. We will move the training time comparison from the appendix to the main text, fix Line 79 in the Introduction, clarify the meaning of $\theta_0 - \theta_R$​ in the caption of Figure 1, and correct the grammar error on Line 390.

---

### Official Review · Reviewer_J3K4 · 2025-03-15

**Overall Recommendation:** 4

**Summary:**

The authors introduce a new LLM unlearning task called Tool Unlearning, designed to remove previously learned tools from tool-augmented LLMs. To tackle this challenge, they develop TOOLDELETE, which incorporates three key properties: tool knowledge deletion, tool knowledge retention, and general capability retention. Extensive experiments across multiple benchmarks validate the effectiveness of TOOLDELETE.

**Claims And Evidence:**

The authors introduce a new task for LLM unlearning and highlight its significance. To address this challenge, they present TOOLDELETE as a solution. While they do not provide a theoretical analysis, their extensive experimental results effectively support their claims.

**Essential References Not Discussed:**

The authors have cited most related works.

**Experimental Designs Or Analyses:**

The authors present detailed experimental settings, evaluations, and baselines. The design and analysis are well-structured and valid.

**Methods And Evaluation Criteria:**

The authors compare the proposed TOOLDELETE with several baselines, including general and LLM-specific unlearning approaches, and conduct extensive experiments on widely used benchmarks such as ToolAlpca, ToolBench, and API-Bench.

**Other Comments Or Suggestions:**

See the above section.

**Other Strengths And Weaknesses:**

Strengths

1. The authors clearly define the new task and highlight three key properties essential for addressing it: tool knowledge deletion, tool knowledge retention, and general capability retention. They effectively illustrate both the problem and their proposed solution in Figure 1.
2. The introduction of LiRA-Tool, a membership inference attack (MIA) specifically designed for tool unlearning, provides an effective metric to assess whether tool-related knowledge has been successfully removed.
3. The authors propose the use of "shadow samples" as a strategy to remove the dependency on accessing training data.
4. Unlike many traditional methods, TOOLDELETE can support sequential unlearning efficiently.

Weaknesses

1. While the empirical results are strong, the paper does not provide a formal theoretical foundation to support its claims about unlearning effectiveness.
2. Although the authors introduce TOOLDELETE, it primarily builds on existing methods and lacks significant technical novelty.

**Questions For Authors:**

1. What do you mean by this "By moving beyond limited training prompts, LiRA-Tool ensures that the model loss reflect overall tool-using ability, rather than just sample level memorization."
2. What are the test sets $T_T$ used in the experiments?

**Relation To Broader Scientific Literature:**

This paper introduces a novel unlearning task that contributes to the research on LLM unlearning.

**Theoretical Claims:**

The authors do not include a theoretical analysis.

---

> ### Author Rebuttal · Authors · 2025-04-01
>
> **Re W1: While the empirical results are strong, the paper does not provide a formal theoretical foundation to support its claims about unlearning effectiveness.**
>
> Thank you for highlighting this important point. We acknowledge that the paper does not include a formal theoretical framework. Our primary focus in this work is on addressing the practical challenges of tool unlearning in tool-augmented LLMs—a new task that has not been investigated before. We believe our empirical results, which span multiple datasets, robust evaluation settings, and ablation studies provide strong evidence for the effectiveness of ToolDelete.
>
>
> **Re W2: Although the authors introduce TOOLDELETE, it primarily builds on existing methods and lacks significant technical novelty.**
>
> Thank you for the feedback. The key novelty of ToolDelete is in integrating targeted forgetting and general utility preservation. ToolDelete addresses several challenges such as removing abstract and distributed parametric knowledge of tools, and preserving general utility for targeted forgetting--that challenges have not been addressed by existing sample-level unlearning methods. In addition, the paper also introduces LiRA-Tool, the first evaluation method for assessing tool-level knowledge removal. We believe these contributions present a practical and technically meaningful contribution to existing literature in unlearning.
>
>
>
> **Re Q1: Please clarify what the authors mean by the following statement: "By moving beyond limited training prompts, LiRA-Tool ensures that the model loss reflect overall tool-using ability, rather than just sample level memorization."**
>
> Traditional MIA methods like LiRA aim to determine if specific samples were part of the training data. However, in tool unlearning, our goal is to assess whether the model has truly forgotten how to use a tool, which requires evaluating its broader functional capability, not just whether it "remembers" certain samples. By "moving beyond limited training prompts," we mean that relying solely on the original training data (which may be narrow in scope or limited in diversity) could lead to incomplete or misleading conclusions. Instead, LiRA-Tool uses diverse shadow prompts—with variations in format, intent, and difficulty—to probe different aspects of a tool’s usage. This allows us to stress-test the model and evaluate whether the tool-related knowledge is removed from its parametric behavior, rather than checking for surface-level memorization. This is in particular important as LLM unlearning is susceptible to re-learning attacks (Hu et al., ICLR 2025) or jailbreak attacks (Lynch et al., 2024). LiRA-Tool provides a more fundamental evaluation under these cases.
> [1] Unlearning or Obfuscating? Jogging the Memory of Unlearned LLMs via Benign Relearning. Shengyuan Hu, Yiwei Fu, Zhiwei Steven Wu, Virginia Smith, ICLR 2025
>
> [2] EIGHT METHODS TO EVALUATE ROBUST UNLEARNING IN LLMS. Aengus Lynch, Phillip Guo, Aidan Ewart, Stephen Casper, 2024
>
>
> **Re Q2: What are the test sets T_T used in the experiments?**
>
> These are the official test sets that were originally provided for each dataset. We used the same splits for consistency purposes.

---

### Decision · Program_Chairs · 2025-05-01

**Decision:**

Accept (poster)

**Comment:**

This paper studies machine unlearning for tool-augmented LLMs by proposing TOOLDELETE, a method designed to remove previously learned tools. TOOLDELETE satisfies three key properties: tool knowledge removal, tool knowledge retention, and general capability retention. The paper also introduces an adapted membership inference attack (MIA) method to evaluate the effectiveness of tool unlearning.

The reviewers appreciate the following strengths:
1) The paper addresses an emerging and important task—machine unlearning for tool-augmented LLMs.
2) It proposes a new tool unlearning method with a clear definition of desiderata.
3) It adapts the LiRA-Tool membership inference attack (MIA) as an evaluation method for assessing the sufficiency of tool unlearning.
4) The paper is overall well-written and well-structured.

However, the reviewers also raised some concerns:
1) A lack of theoretical analysis regarding tool-unlearning effectiveness.
2) TOOLDELETE incorporates some techniques from existing LLM unlearning literature.
3) Insufficient clarity in the experimental settings.
4) A limited introduction and comparison with existing MIA approaches.

During the discussion, the authors clarified the new technical contributions, experimental settings, and the motivation and novelty of LiRA-Tool, as well as providing new experimental results.

Three reviewers are positive following the discussion. One reviewer remains negative, primarily due to concerns that TOOLDELETE borrows some techniques (e.g., the unlearning objective function) from existing LLM unlearning methods. Regarding novelty, the area chair aligns with the majority of reviewers and considers the formulation and solution of the tool-unlearning task to be novel. Overall, the paper is at the borderline for acceptance. Therefore, the area chair recommends weak acceptance of the paper, with the final decision left to the SAC and PCs.